# RoVi-Aug: Robot and Viewpoint Augmentation for Cross-Embodiment Robot Learning

**Lawrence Yunliang Chen**[*1], **Chenfeng Xu**[*1], **Karthik Dharmarajan**[1],
**Richard Cheng**[2], **Kurt Keutzer**[1], **Masayoshi Tomizuka**[1],
**Quan Vuong**[3], **Ken Goldberg**[1]
[1]UC Berkeley  [2]Toyota Research Institute  [3]Physical Intelligence

**Abstract:** Scaling up robot learning requires large and diverse datasets, and how to efficiently reuse collected data and transfer policies to new embodiments remains an open question. Emerging research such as the Open-X Embodiment (OXE) project has shown promise in leveraging skills by combining datasets including different robots. However, imbalances in the distribution of robot types and camera angles in many datasets make policies prone to overfit. To mitigate this issue, we propose RoVi-Aug, which leverages state-of-the-art image-to-image generative models to augment robot data by synthesizing demonstrations with different robots and camera views. Through extensive physical experiments, we show that, by training on robot- and viewpoint-augmented data, RoVi-Aug can zero-shot deploy on an unseen robot with significantly different camera angles. Compared to test-time adaptation algorithms such as Mirage, RoVi-Aug requires no extra processing at test time, does not assume known camera angles, and allows policy fine-tuning. Moreover, by co-training on both the original and augmented robot datasets, RoVi-Aug can learn multi-robot and multi-task policies, enabling more efficient transfer between robots and skills and improving success rates by up to 30%. Project website: https://rovi-aug.github.io.

**Keywords:** Cross-Embodiment Learning, Viewpoint Robust, Data Augmentation

## 1 Introduction

Emerging research in robot learning suggests that scaling up data can help learned policies be more generalizable and robust [1, 2, 3, 4, 5, 6, 7, 8, 9]. However, compared to state-of-the-art foundation models [10] in computer vision (CV) [11, 12, 13, 14] and natural language processing (NLP), the size of robotic data is still several orders of magnitude smaller than those used to train large language and multi-modal models [15, 16, 17, 18, 19]. Collecting real robot data is time-consuming [20, 21, 22] and labor intensive [2, 3, 5, 23, 24], and ensuring data diversity for generalizable policies requires careful balance [25]. Can we more effectively leverage currently available real robot data?

In an unprecedented community effort, the Open-X Embodiment (OXE) project [9] combines 60 robot datasets and finds that co-training can exhibit positive transfer and improves the capabilities of multiple robots by leveraging experience from each other. However, the OXE dataset is highly unbalanced, dominated by a few robot types such as Franka and xArm. Additionally, most datasets have a limited diversity of camera poses. Policies trained on such data tend to overfit to those robot types and viewpoints and need fine-tuning when deploying on other robots or at even slightly different camera angles. To mitigate this issue, a test-time adaptation algorithm, Mirage [26], uses "cross-painting" to transform an unseen target robot into the source robot seen during training, to create an illusion as if the source robot is performing the task at test time. While Mirage can achieve zero-shot transfer on unseen target robots, it has a few limitations: (1) It requires precise robot models and camera matrices; (2) It does not allow policy finetuning; (3) It is limited to small camera pose changes due to depth reprojection error.

8th Conference on Robot Learning (CoRL 2024), Munich, Germany.

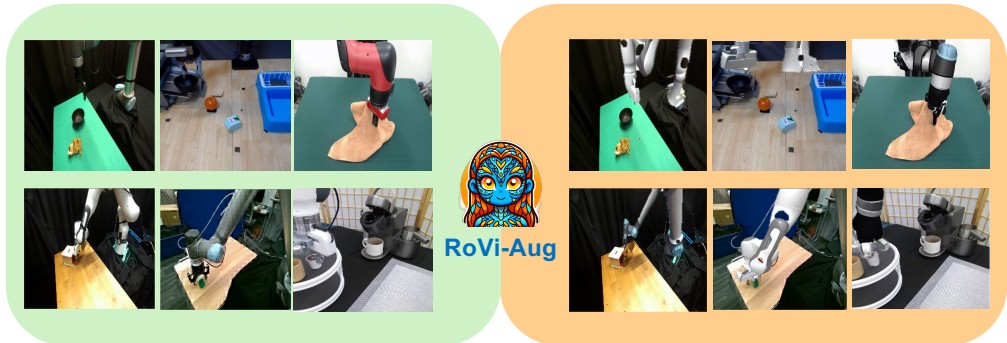

Figure 1: Given robot images, RoVi-Aug uses state-of-the-art diffusion models to augment the data and generate synthetic images with different robots and viewpoints. Policy trained on the augmented dataset can be deployed on the target robots zero-shot or further finetuned, exhibiting robustness to camera pose changes.

In this work, we seek to bridge these limitations. Rather than naively co-training on combined data from multiple robots, we aim to more explicitly encourage the model to learn the cross-product of the robots and skills contained in each dataset. We aim to improve the robustness and generalizability of the policy to different robot visuals and camera poses during training instead of relying on an accurate test-time cross-painting pipeline. We propose RoVi-Aug, a robot and viewpoint augmentation pipeline that synthetically generates images with different robot types and camera poses using diffusion models. Through extensive real-world experiments, we show that, by training on robot- and viewpoint-augmented data, RoVi-Aug can zero-shot control different robots with significantly different camera poses compared to the poses seen during training. In contrast to Mirage, RoVi-Aug does not assume known camera matrices and allows policy fine-tuning to increase performance on challenging tasks. Furthermore, by co-training on original and augmented robot datasets, RoVi-Aug can learn multi-robot and multi-task policies and improve finetuning sample efficiency.

This paper makes 3 contributions:

1. RoVi-Aug, a novel approach to robot data augmentation that uses diffusion models to generate trajectories with novel robots and viewpoints;

2. Physical experiments with Franka and UR5 suggesting that robot augmentation enables zero-shot deployment on target robots and viewpoint augmentation improves the robustness of policies to camera pose changes. When combined, they yield policies that work for target robots at camera poses significantly different from those in the initial demonstration data;

3. Experiments suggesting that RoVi-Aug can learn multi-robot multi-task policies and improve the finetuning sample efficiency of a generalist policy on novel robot-task combinations.

## 2   Related Work

### 2.1   Cross-Embodiment Robot Learning

Recognizing the high cost of collecting real robot data, many prior works have studied using other data sources, such as simulation [27, 28, 29, 30, 31, 32], other robot data [33, 34], and human or animal videos [35, 36, 37, 38, 39, 40, 41, 35, 42, 43, 44, 45, 46, 47, 48], to increase sample efficiency and accelerate learning [49]. In a transfer learning setting, one can first pretrain a visual encoder [50], dynamics model [51], or policy [52, 53, 54] and then perform online finetuning using reinforcement learning. In a cross-domain imitation paradigm, methods often involve learning correspondences between the source and target domains [55, 56, 57, 58], and then constructing auxiliary rewards [59, 55, 60, 61] or applying adversarial training [62, 63, 64]. Ghadirzadeh et al. [65] use meta-learning to enable a new robot to quickly learn from few-shot trajectories at test time.

Cross-embodiment learning could also be used to learn more robust and generalizable policies through joint training in a multi-robot multi-task fashion. For example, by training on a family of robots with varying kinematics and dynamics in simulation, robot-conditioned policies [66, 33, 67, 68, 69, 70, 71, 72, 73, 74] are robust to novel morphologies within the range of training

distribution, and modular policies [1, 75, 76, 77, 78] can be more transferrable to different robots and tasks. More recently, many works have also explored training on large and diverse real robot data [79, 80, 81, 82, 24, 23, 83, 84] to learn visual representations [85, 86, 87] and predictive world models [88, 89, 90] and showed that policies trained are more generalizable to new objects, scenes, tasks, and embodiments [2, 3, 4, 6, 7, 8, 91, 92, 93, 94, 95, 96, 97, 98, 99]. In this work, we build on these insights and propose to more explicitly encourage positive transfer between robots and skills by performing data augmentation.

Our method is inspired by Mirage [26], a recent test-time adaptation algorithm that uses "cross-painting" to achieve cross-embodiment policy transfer by replacing the target robot in the image with a source robot seen during training. While Mirage avoids modifying the source robot policy and enables zero-shot transfer, it has several limitations, such as requiring a fast renderer, precise robot models, and accurate camera calibration. We address these issues by using training time data augmentation with diffusion models trained on randomized robot poses and camera angles, eliminating the need for camera matrix knowledge. Our approach additionally allows zero-shot deployment as well as finetuning or cotraining on additional data to improve the performance and learn multi-robot multi-skill policies that are robust to significant camera angle changes.

## 2.2 Generative Models and Data Augmentation in Robotics

With the significant progress in generative models including large language and multi-modal models [15, 16, 99, 100] and diffusion models [101, 11, 102, 103] trained on Internet-scale data, there is a growing interest in leveraging these models for robotics. For example, prior work has explored using language models for planning [104, 105, 106, 107, 108], control [109], reward specification [110, 111], and data relabeling [112]. Image and video generation models have been used for generative simulation [113, 114], data augmentation [115, 116, 117, 97] and visual goal planning [89, 118]. Our method falls into the data augmentation category. However, unlike prior work that generates distractor objects, backgrounds, and new tasks [115, 116, 117, 97], we use diffusion models to generate alternative robots and camera viewpoints. As such, RoVi-Aug enables trained policies to generalize to different robots with different camera setups.

## 2.3 Viewpoint Adaptation and Viewpoint Robust Policy

Visuomotor control policies that take in images as inputs tend to overfit to the camera angle in the training data, and even small changes between training and testing could severely hurt performance [26, 119]. While using 3D representations [94, 120] alleviates the problem, it requires a calibrated depth camera or multiple views [94, 121], and is more computationally expensive. For mobile robots, Hirose et al. [122] extract a 3D point cloud from the training data and performs re-rendering, and Ex-DoF [123] applies virtual rotation of the robot's 360° camera to augment training data. To improve viewpoint robustness of image-based policies, Sadeghi et al. [119] use a recurrent neural network to understand how actions affect arm movement through history. Seo et al. [124] use many simulated viewpoints to learn a visual representation, whose downstream policy exhibits viewpoint robustness. Instead of pretraining in simulation with diverse rendering, we synthesize novel views of real scenes. SPARTN [125] and DMD [126] use neural radiance fields (NeRFs) and diffusion models, respectively, to generate perturbed viewpoints for wrist cameras, whereas our viewpoint augmentation applies to fixed third-person views.

## 3 Problem Statement

We assume a demonstration dataset $\mathcal{D}^{\mathcal{S}} = \{\tau_1^{\mathcal{S}}, \tau_2^{\mathcal{S}}, ..., \tau_n^{\mathcal{S}}\}$ consisting of $n$ successful trajectories of a source robot $\mathcal{S}$ performing some task. Each trajectory $\tau_i^{\mathcal{S}} = (\{o_{1..H_i}^{\mathcal{S}}\}, \{p_{1..H_i}^{\mathcal{S}}\}, \{a_{1..H_i}^{\mathcal{S}}\})$, where $\{o_1^{\mathcal{S}}, ..., o_{H_i}^{\mathcal{S}}\}$ is a sequence of RGB camera observations, $\{p_1^{\mathcal{S}}, ..., p_{H_i}^{\mathcal{S}}\}$ is the sequence of corresponding gripper poses, and $\{a_1^{\mathcal{S}}, ..., a_{H_i}^{\mathcal{S}}\}$ is the sequence of corresponding robot actions. This dataset can be used to train models with behavior cloning for robot $\mathcal{S}$. Our goal is to augment $\mathcal{D}^{\mathcal{S}}$ into $\mathcal{D}^{\text{Aug}}$ such that we can learn a policy that can be successfully deployed on a different robot $\mathcal{T}$, known as the target robot, with a potentially different camera viewpoint. In this work, we focus on robot arms mounted on a stationary base and assume the grippers are similar in shape and function.

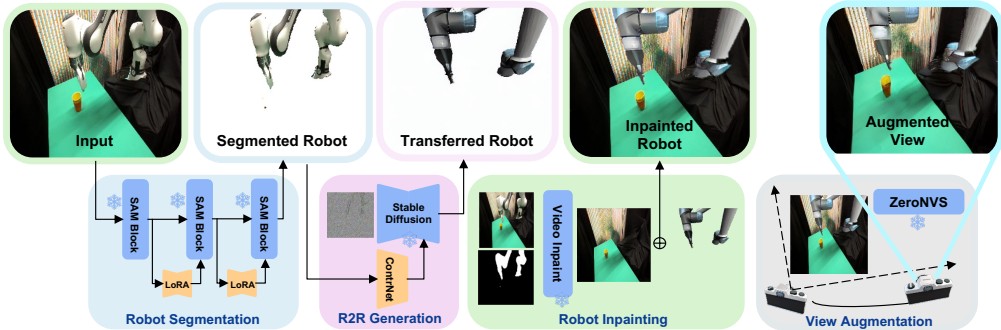

Figure 2: **Overview of the RoVi-Aug pipeline.** Given an input robot image, we first segment the robot out using a finetuned SAM [129] model, then use a ControlNet [130] to transform the robot into another robot. After pasting the synthetic robot back into the background, we use ZeroNVS [131] to generate novel views.

Similar to prior work [7, 26, 127, 128], we use Cartesian control and assume knowledge of the two robots' end effector coordinate frames with respect to their bases (e.g., moving forward corresponds to an increase in the $x$-axis) such that we can use a rigid transformation $T_{\mathcal{T}}^{\mathcal{S}}$ to preprocess the data and align the robots' end effector poses $p^{\mathcal{S}} = T_{\mathcal{T}}^{\mathcal{S}} p^{\mathcal{T}}$ and actions $a^{\mathcal{S}} = T_{\mathcal{T}}^{\mathcal{S}} a^{\mathcal{T}}$ into the same vector space. Thus, for notational convenience, we omit the superscript differentiating gripper poses and actions between the robots. However, the image observations $o^{\mathcal{S}}$ and $p^{\mathcal{T}}$ cannot be easily aligned since the robots may look very different. We do not assume knowledge of the camera matrices in either setup.

After augmentation, we learn a policy $\pi(a_t | o_t^{\mathcal{T}}, p_t)$ on $\mathcal{D}^{\mathrm{Aug}}$ using imitation learning. At test time, it takes as inputs the observations from the target robot and outputs actions that can be deployed on the target robot. Additionally, by co-training on the original data $\mathcal{D}^{\mathcal{S}}$ as well as $\mathcal{D}^{\mathrm{Aug}}$, we can also obtain a multi-robot policy.

# 4 RoVi-Aug

In this section, we describe RoVi-Aug, an automated pipeline for augmenting and scaling up robot data. Our key insight is that the robot's actions should be invariant to its visual appearances and camera viewpoints. Our robot augmentation pipeline leverages state-of-the-art diffusion models [11, 129] to synthesize alternative robots and novel viewpoints. Fig. 2 illustrates RoVi-Aug pipeline.

## 4.1 Robot Augmentation (Ro-Aug)

Given a sequence of robot image observations $D_i^{\mathcal{S}} = \{o_1^{\mathcal{S}}, ..., o_{H_i}^{\mathcal{S}}\}$, we seek to transform the robot $\mathcal{S}$ in the images into a different robot $\mathcal{T}$ at the same gripper pose, a process known as cross-painting. While Mirage [26] proposes to perform cross-painting using a renderer to compute source robot masks and target robot visuals, it requires precise camera calibration which is unavailable for most open-source datasets. To relax this assumption, we approach cross-painting as an image-to-image translation problem. RoVi-Aug begins by predicting semantic mask on the robot $\mathcal{S}$, which are then extracted and transformed into robot $\mathcal{T}$ using a robot-to-robot (R2R) diffusion model. Meanwhile, the masked regions in the original images are inpainted using a video inpainting network to ensure visual continuity and integrity. Finally, the generated robot $\mathcal{T}$ is pasted back into the background image (see Fig. 2).

**Robot Segmentation.** In order to replace robot $\mathcal{S}$ with robot $\mathcal{T}$ in the image, we first need to detect the robot using semantic segmentation [132, 129, 133, 134]. We find that off-the-shelf segmentation models [129, 135] often fail to accurately segment out the robot, potentially due to the fact that robot images are under-represented in their training data. As such, we finetune a pretrained Segment Anything Model (SAM) [129] using Low-Rank Adaptation (LoRA) [136]. We use simulation to synthetically generate a large dataset of different robot images with corresponding masks, where we randomly sample a wide range of camera and robot poses. We apply brightness augmentation and resizing to simulate different lighting and fields of view. To create diverse backgrounds, we paste the generated robot parts into various background images [137]. By training the LoRA layer on this synthetic dataset, we obtain a mask model capable of handling different robot and camera poses.

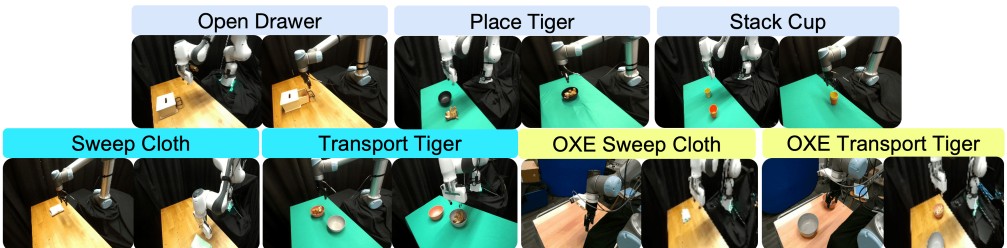

Figure 3: **Tasks used for evaluation.** For each task, on the left is an example training view and robot, and on the right is the different test-time embodiment.

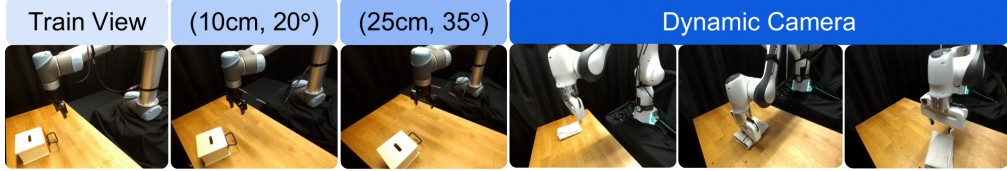

Figure 4: **Evaluated camera views.** For static third-person cameras, we perturb the initial training view by 10 cm translation, 20° rotation and 25 cm translation, 35° rotation. Even when the camera is moving dynamically, RoVi-Aug is able to successfully sweep the cloth.

**Robot-to-Robot (R2R) Generation** Next, we aim to transform the segmented robot $\mathcal{S}$ into robot $\mathcal{T}$. We use an image-to-image diffusion model. Similar to semantic segmentation, training a diffusion model capable of handling various camera and robot poses requires a large dataset of paired images. As collecting paired real robot data is challenging due to the need for precise adjustments of camera and robot poses, we again use simulation to generate pairs of robots at the same randomly sampled robot poses and camera poses, with brightness and resizing augmentations. Inspired by [138, 130], we use a ControlNet [130] to finetune a pretrained Stable Diffusion [11]. Even though we train the model on simulation images, we find that it still performs well on real segmented robot images.

**Robot Inpainting** Inspired by Li et al. [139], after segmenting out robot $\mathcal{S}$ from the image, we inpaint the missing region using a video inpainting model E²FGVI [140]. The final step involves pasting the generated robot $\mathcal{T}$ back to the image. As the R2R diffusion model is trained on simulated robot images, there is a visual gap from the real robot, particularly with the illumination. To prevent the trained policy on the augmented data from overfitting to the synthetic robot visuals, we perform random brightness augmentation to the generated robot before pasting it. We find in our experiments that this randomization significantly helps the performance of the trained policy (Section 5.3).

At the end of the Robot Augmentation pipeline, we obtain a sequence of cross-painted observations with synthesized target robot: $D_i^{\mathcal{S}\rightarrow\mathcal{T}} = \{o_1^{\mathcal{S}\rightarrow\mathcal{T}}, ..., o_{H_i}^{\mathcal{S}\rightarrow\mathcal{T}}\}$.

### 4.2 Viewpoint Augmentation (Vi-Aug)

To increase robustness of the trained policy to camera pose changes, we propose to augment the viewpoints of the images. This is orthogonal to robot augmentation and can be applied to both $D_i^{\mathcal{S}}$ and $D_i^{\mathcal{S}\rightarrow\mathcal{T}}$.

We use ZeroNVS [131], a state-of-the-art 3D-aware diffusion model that can zero-shot synthesize 360° view of a scene from a single image. Compared to prior methods [103, 102, 141] that are limited to segmented object with no background, ZeroNVS works with multi-object scenes with complex backgrounds. For each image $o_t \in D_i$, we uniformly sample perturbations $(\tilde{R}_t, \tilde{T}_t) \in SE(3)$ from a box range, where each component in $\tilde{T}_t$ is bounded by an interval. We parametrize $\tilde{R}_t$ with Euler angles and each of those three angles is uniformly sampled within an interval described in Section 5.1. This process produces a resulting image as if the camera were perturbed by the sampled transformation: $o_t^{\tilde{R},\tilde{T}} = f\left(o_t; \tilde{R}, \tilde{T}\right)$, where we use $f$ to denote the camera transformation. We denote the resulting augmented data as $D_i^{\text{Vi-Aug}} = \{o_1^{R_1,T_1}, ..., o_{H_i}^{R_{H_i},T_{H_i}}\}$. We experiment with two strategies for sampling the perturbations: independently sampling random $(\tilde{R}_t, \tilde{T}_t)$ for each image, or applying a consistent random transformation $(\tilde{R}, \tilde{T})$ across the entire trajectory in $D_i$.

## 4.3 Policy Training

After applying robot and viewpoint augmentation, we can train a policy $\pi$ based on the Diffusion Policy architecture [142] on the augmented dataset $\mathcal{D}^{\mathcal{S}\to\mathcal{T}\,\text{Vi-Aug}}$ and zero-shot deploy the policy on the target robot $\mathcal{T}$. For challenging tasks or when there is a large difference in the dynamics between the robots, we can also collect a small demonstration dataset $\mathcal{D}^{\mathcal{T}}$ on the target robot directly and few-shot finetune $\pi$ on $\mathcal{D}^{\mathcal{T}}$ to further improve policy performance. Alternatively, we can co-train $\pi$ on $\mathcal{D}^{\mathcal{S}\,\text{Vi-Aug}}\bigcup\mathcal{D}^{\mathcal{S}\to\mathcal{T}\,\text{Vi-Aug}}$ to obtain a multi-robot policy. Additionally, if we have multiple datasets with different tasks, can mix-and-match the datasets and train a multi-robot multi-task policy. For example, given data $\mathcal{D}_1^{\mathcal{S}}$ and $\mathcal{D}_2^{\mathcal{T}}$ with robot $\mathcal{S}$ performing task 1 and robot $\mathcal{T}$ performing task 2, we can train on the cross-product $\mathcal{D}_1^{\mathcal{S}}\bigcup\mathcal{D}_2^{\mathcal{T}\to\mathcal{S}}\bigcup\mathcal{D}_1^{\mathcal{S}\to\mathcal{T}}\bigcup\mathcal{D}_2^{\mathcal{T}}$ and their viewpoint-augmented versions to obtain a policy that can perform both tasks on both robots. In this way, we efficiently reuse the datasets and explicitly encourage transfer between robots and skills.

# 5 Experiments

## 5.1 Implementation Details

To train our robot segmentation and Robot-to-Robot generation models, we use the Robosuite simulator [143] to generate a large dataset of paired robot images with corresponding masks with randomly sampled robot poses and camera poses (see supplementary material for details). We use 4 robots: Franka, UR5, Sawyer, and Jaco, with 800k images each. We finetune a LoRA layer while keeping SAM frozen with a learning rate of 1e-4 for just one epoch to avoid the overfitting. We train a ControlNet for each robot pair based on Stable Diffusion v1.5 [11] with a learning rate of 1e-4 for 20k steps. During robot inpainting, we randomly sample perturbations of the value channel in the HSV space between -30 and 30.

For view augmentation sampling, $\tilde{T}_x, \tilde{T}_z \in (-0.25\,\text{m}, 0.25\,\text{m})$, $\tilde{T}_y \in (-0.1\,\text{m}, 0.1\,\text{m})$. The $y$ (vertical) direction has a lower translation range, as we have noticed that when moving excessively along the vertical direction, ZeroNVS outputs larger, more distracting artifacts. For rotation, we sample each Euler angle between $\pm 0.1$ radians.

## 5.2 Experiment Setup

We design experiments to answer the following research questions: (1) Can robot augmentation (Ro-Aug) effectively bridge the visual gap between the robots? (2) Can viewpoint augmentation (Vi-Aug) improve policy robustness to camera pose changes? (3) Can policies trained with RoVi-Aug be successfully deployed zero-shot on a different robot with camera changes? (4) Does RoVi-Aug enable multi-robot multi-task training and better facilitate transfer between robots and skills?

| Tasks
Policies | Franka → UR5 | | | UR5 → Franka | |
|---|---|---|---|---|---|
| | Open Drawer | Place Tiger | Stack Cup | Sweep Cloth | Transport Tiger |
| No Augmentation | 0% | 0% | 0% | 0% | 40% |
| Mirage | 60% | **90%** | **50%** | **100%** | 70% |
| Ro-Aug | **90%** | 80% | 30% | **100%** | **80%** |
| Ro-Aug w/o Bright. Rand. | **90%** | 50% | 10% | 40% | 60% |

Table 1: **Zero-shot physical experiments evaluating robot augmentation.** We evaluate Ro-Aug on 5 tasks in 2 settings with 10 trials each: Learning a policy using Franka demonstration data and evaluating on a UR5, and vice versa. The camera poses are the same. We compare Ro-Aug with 2 baselines and an ablation that does not apply random brightness augmentation during the Ro-Aug pipeline. We see that Ro-Aug achieves comparable zero-shot performance as Mirage.

To answer the first three questions, we study policy transfer between a Franka and a UR5 robot on 5 tasks (Fig. 3): (1) Open a drawer, (2) Pick up a toy tiger from the table and put it into a bowl (Place Tiger), (3) Stack cups, (4) Sweep cloth from right to left, and (5) Transport a toy tiger between two bowls. See the Appendix for more details. For the first three tasks, we collect demonstrations on the Franka, and for the latter two, we collect demonstrations on the UR5. All demonstrations are collected via teleoperation at 15 Hz [5], with 150 trajectories each. A typical trajectory consists

| Tasks | Franka → UR5 | | | UR5 → Franka | |
|---|---|---|---|---|---|
| Policies | Open Drawer | Place Tiger | Stack Cup | Sweep Cloth | Transport Tiger |
| 5-Shot | 40% | 30% | 0% | 50% | 40% |
| Ro-Aug + 5-Shot | **100%** | **100%** | **60%** | **100%** | **100%** |
| 10-Shot | 70% | 40% | 50% | 80% | 80% |
| Ro-Aug + 10-Shot | **100%** | **100%** | **80%** | **100%** | **100%** |

Table 2: **Few-shot physical experiments evaluating robot augmentation.** We apply 5-shot and 10-shot finetuning to policies trained with Ro-Aug and compare them to few-shot policies trained without Ro-Aug. We see that Ro-Aug improves finetuning sample efficiency and exceeds the performance of all policies in Table 1.

| Tasks | Place Tiger (Franka → Franka) | | | |
|---|---|---|---|---|
| Policies | Same Angle | 10 cm, 20° | 25 cm, 35° | 35 cm, 45° |
| No Augmentation | **100%** | 0% | 0% | 0% |
| Vi-Aug 10 cm - Consistent | **100%** | 30% | 0% | 0% |
| Vi-Aug 10 cm - Inconsistent | **100%** | 70% | 10% | 0% |
| Vi-Aug 25 cm - Consistent | **100%** | **80%** | 30% | 30% |
| Vi-Aug 25 cm - Inconsistent | 90% | **80%** | 50% | 30% |
| Vi-Aug 40 cm - Consistent | 70% | 70% | **60%** | 20% |
| Vi-Aug 40 cm - Inconsistent | 80% | **80%** | 50% | **40%** |

Table 3: **Physical experiments evaluating viewpoint augmentation.** We compare policies trained with different degrees of camera perturbations (rows). The numbers represent the range of the camera perturbation that $\tilde{T}_x$ and $\tilde{T}_z$ are sampled from. "Consistent/Inconsistent" represents whether the same/different perturbation is applied to each timestep in a trajectory. We evaluate on the same robot but with different test camera angles (columns).

of 75-120 timesteps (5-8 s). We use a ZED 2 camera positioned from the side for each robot. We augment the demonstration data with robot augmentation (Ro-Aug) using the other robot, viewpoint augmentation (Vi-Aug), as well as both (RoVi-Aug), train a diffusion policy, and evaluate on the other robot. All experiments are evaluated with 10 trials each.

To answer the last question, we combine demonstration data from Franka and UR5 for different tasks, perform robot augmentations, and train a multi-robot multi-task diffusion policy. We also select the Berkeley UR5 dataset [144] from the OXE data [9], apply RoVi-Aug to generate synthetic Franka images and finetune a generalist policy, Octo [145], on the augmented datasets. We additionally collect 50 demonstrations on the target robot (Franka) and further finetune Octo-Base in a language goal-conditioned format. We compare whether training Octo on the augmented data improves the finetuning sample efficiency on the downstream tasks.

## 5.3 Results

Table 1 shows the effect of robot augmentation when the camera poses are the same. The policy is deployed zero-shot. We compare Ro-Aug with 2 baselines, no augmentation and Mirage, and an ablation that does not apply random brightness augmentation during the Ro-Aug pipeline. Without robot augmentation, the policy trained on the source robot only barely achieves success on the target robot. On the other hand, Ro-Aug achieves comparable zero-shot performance as Mirage. Additionally, we see that brightness randomization helps performance, suggesting that it effectively prevents the policy from overfitting to the lighting in simulation that the R2R model is trained on.

Table 2 shows the policies trained on Ro-Aug data can be finetuned with 5-10 demonstrations on the target robot to further improve performance. Compared to few-shot policies trained without Ro-Aug, we see that Ro-Aug improves finetuning sample efficiency and exceeds the performance of all policies in Table 1. In contrast, Mirage does not allow finetuning and cannot improve performance on challenging tasks such as cup stacking.

Table 3 evaluates the effect of viewpoint augmentation. We choose the Tiger Place task on the Franka robot and study how different strategies of camera perturbation sampling affect policy robustness. We sample translations $\tilde{T}_x$ and $\tilde{T}_z$ between $\pm 0.1$ m, $\pm 0.25$ m, and $\pm 0.4$ m, and compare consistent perturbation across trajectories or independently on each image. From Table 3, we see that larger variation during augmentation improves policy robustness under severe camera pose changes.

However, the performance decreases under the original camera angle, potentially due to lower density of each camera pose as the sampling range increases. Additionally, inconsistent augmentation seems to slightly outperform consistent augmentation., suggesting potential benefit from more augmentation. We note that the diffusion policy takes in only 2 steps of history, so viewpoint inconsistency may not matter much. Future work can study whether inconsistent augmentation would harm policies that use a longer history. Based on the results, we choose to apply inconsistent augmentation with 25 cm perturbation range for other RoVi-Aug experiments.

| Tasks Policies | Franka → UR5 | | | | UR5 → Franka | | | |
| | Open Drawer | | Place Tiger | | Sweep Cloth | | Transport Tiger | |
| | 10 cm, 20° | 25 cm, 35° | 10 cm, 20° | 25 cm, 35° | 10 cm, 20° | 25 cm, 35° | 10 cm, 20° | 25 cm, 35° |
|---|---|---|---|---|---|---|---|---|
| Mirage | 50% | 30% | 30% | 20% | **80%** | 30% | 20% | 0% |
| Ro-Aug | 60% | 20% | 30% | 10% | 0% | 0% | 0% | 0% |
| RoVi-Aug | **80%** | **50%** | **70%** | **30%** | **80%** | **40%** | **40%** | **30%** |

Table 4: **Physical experiments evaluating RoVi-Aug on different robots with different camera angles.** The translation and rotation shows the difference in the camera poses between the robots. Mirage uses a policy trained on only the source robot with a test-time cross-painting procedure and depth reprojection to account for camera pose changes. Ro-Aug only applies robot augmentation while RoVi-Aug applies both robot and viewpoint augmentation. For both Ro-Aug and RoVi-Aug, the policy is trained on the augmented data and deployed on the target robot zero-shot.

| | Franka | UR5 |
|---|---|---|
| Place Tiger | 80% | 70% |
| Transport Tiger | 60% | 80% |

Table 5: **Robot-Skill Cross Product.** We train a multi-robot multi-task diffusion policy trained on pooling the Franka Tiger Place data and UR5 Tiger Transport data as well as their RoVi-Aug versions together.

| Tasks Policies | OXE UR5 → Franka | |
| | Sweep Cloth | Transport Tiger |
|---|---|---|
| Octo-Base | 30% | 20% |
| Octo-Base + RoVi-Aug | **60%** | **40%** |

Table 6: **Octo finetuning from the OXE datasets with 50 in-domain demonstrations for each task.** RoVi-Aug improves finetuning sample efficiency.

Table 4 evaluates RoVi-Aug on different robots with different viewpoints. We can see that viewpoint augmentation is crucial and Mirage struggles with larger camera pose changes. In contrast, RoVi-Aug can still achieve success when the target robot viewpoint is significantly different from source robot.

To evaluate robot-skill cross-product, we combine the Tiger Place demonstration data from the Franka and Tiger Transport demonstration data from the UR5, as well as their robot-augmented UR5 and Franka versions, and train a multi-robot multi-task diffusion policy. From Table 5, we can see that the policy can successfully execute the two tasks on both robots. Additionally, we evaluate whether RoVi-Aug improves finetuning sample efficiency. From Table 6, we can see that after training Octo on the augmented OXE data, the policy has seen the synthetic target robots performing the tasks, accelerating downstream finetuning of similar tasks.

## 6 Limitations and Future Work

We present RoVi-Aug, a pipeline for robot and viewpoint augmentation that bridges different robot datasets and better facilitates transfers between robots and skills. There are several limitations, which open up possibilities for future work: (1) Our robot augmentation pipeline relies on a sequence of different models so artifacts can cascade. For example, inaccuracies in the robot segmentation stage (e.g., mistakenly segmenting the object out) could lead to bad robot-to-robot generations in the second stage. See the Appendix for more details on artifacts. Additionally, instead of training an R2R diffusion model for each robot pair, future work could explore a unified model that handles multiple pairs. (2) For viewpoint augmentation, future work could improve the quality of novel view synthesis by finetuning the model on robotics data or using video-based models [146]. (3) While we mitigate viewpoint changes in this work, there are also often background changes in practice during cross-embodiment transfer. Future work could combine RoVi-Aug with prior orthogonal approaches such as object, background, and task augmentation [115, 116] to further obtain more generalizable policies. (4) We only demonstrate transfer between stationary robot arms and do not consider very different grippers such as multi-fingered hands. We leave these extensions to future work.

**Acknowledgments**

This research was performed at the AUTOLab at UC Berkeley in affiliation with the Berkeley AI Research (BAIR) Lab, and the CITRIS "People and Robots" (CPAR) Initiative, and in collaboration with Google DeepMind. The authors are supported in part by donations from Google, Toyota Research Institute, and equipment grants from NVIDIA. L.Y. Chen is supported by the National Science Foundation (NSF) Graduate Research Fellowship Program under Grant No. 2146752. We thank reviewers for valuable feedback.

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

# 7 Appendix

In this section, we provide additional implementation details of RoVi-Aug and our physical experiments.

## 7.1 Algorithm Pseudocode

In this section, we provide the pseudocode for Ro-Aug and Vi-Aug.

---

**Algorithm 1** Ro-Aug

---

**Input:** A sequence of source robot image observations $D_i^{\mathcal{S}} = \{o_1^{\mathcal{S}}, ..., o_{H_i}^{\mathcal{S}}\}$
**Output:** A sequence of cross-painted observations with synthesized target robot: $D_i^{\mathcal{S} \to \mathcal{T}} = \{o_1^{\mathcal{S} \to \mathcal{T}}, ..., o_{H_i}^{\mathcal{S} \to \mathcal{T}}\}$

1: **function** RO-AUG($D_j^{\mathcal{S}}$)
2:      **for** each image $o_j^{\mathcal{S}}$ in $D_i^{\mathcal{S}}$ **do**
3:          Segment the source robot out, resulting in the robot $r_j^{\mathcal{S}}$ and background $b_j^{\mathcal{S}}$ where $o_j^{\mathcal{S}} = r_j^{\mathcal{S}} \cup b_j^{\mathcal{S}}$
4:      **end for**
5:      **for** each robot image $r_j^{\mathcal{S}}$ **do**
6:          Apply the Robot-to-Robot generation model to get $r_j^{\mathcal{S} \to \mathcal{T}}$
7:      **end for**
8:      Apply video inpainting model E2FGV to the background video $\{b_1^{\mathcal{S}}, ..., b_{H_i}^{\mathcal{S}}\}$ to get $\{\tilde{b_1}^{\mathcal{S}}, ..., \tilde{b_{H_i}}^{\mathcal{S}}\}$
9:      **for** each robot image $r_j^{\mathcal{S} \to \mathcal{T}}$ **do**
10:          $o_j^{\mathcal{S} \to \mathcal{T}}$ = Overlay $r_j^{\mathcal{S} \to \mathcal{T}}$ onto $\tilde{b_j}^{\mathcal{S}}$    `#Combine the background and the generated target robot images`
11:      **end for**
12:      **return** $D_i^{\mathcal{S} \to \mathcal{T}} = \{o_1^{\mathcal{S} \to \mathcal{T}}, ..., o_{H_i}^{\mathcal{S} \to \mathcal{T}}\}$
13: **end function**

---

**Algorithm 2** Vi-Aug

---

**Input:** A sequence of robot image observations $D_i = \{o_1, ..., o_{H_i}\}$
**Output:** A sequence of viewpoint augmented images: $D_i^{\text{Vi-Aug}} = \{o_1^{R_1, T_1}, ..., o_{H_i}^{R_{H_i}, T_{H_i}}\}$

1: **function** VI-AUG($D_i$)
2:      **for** each image $o_j$ in $D_i$ **do**
3:          Sample perturbations $(\tilde{R}_j, \tilde{T}_j) \in SE(3)$ from a box range
4:          Generate augmented images using ZeroNVS $f$: $o_j^{\tilde{R}, \tilde{T}} = f\left(o_j; \tilde{R}, \tilde{T}\right)$
5:      **end for**
6:      **return** $D_i^{\text{Vi-Aug}} = \{o_1^{R_1, T_1}, ..., o_{H_i}^{R_{H_i}, T_{H_i}}\}$
7: **end function**

---

## 7.2 Robot Augmentation

### 7.2.1 Training Data Generation

To train our robot segmentation and Robot-to-Robot generation models, we use the Robosuite simulator [143] to generate a large dataset of paired robot images with corresponding masks with randomly sampled robot poses and camera poses. The sampling procedure is as follows: The robot pose is specified by the end-effector pose. The translation component is sampled uniformly with $(x, y, z) \in [-0.25, 0.25] \times [-0.25, 0.25] \times [0.6, 1.3]$ (unit in meters). For the rotation component, we parameterize it as [inward, rightward, z_axis]. To bias the unit vector z_axis towards pointing downward, we parameterize it using spherical coordinate $\theta, \phi$ where $\theta$ (zenith angle) is sampled from a normal distribution $\mathcal{N}(\pi, \pi/3.5)$ and $\phi$ (azimuthal angle) is uniformly sampled between 0 and $2\pi$.

After sampling the robot pose, we randomly sample the camera pose with the following procedure: The position is sampled from a half hemisphere with radius $r \in \mathcal{N}(0.85, 0.2)$ and zenith angle $\theta \in \mathcal{N}(\pi/4, \pi/2.2)$, and azimuthal angle $\phi \in \text{Unif}[-\pi \cdot 3.7/4, \pi \cdot 3.7/4]$. The viewing direction is

towards the center of the hemisphere, which we offset as the gripper position. We also sample camera field of view between 40 and 70. Finally, we randomly perturb the camera pose with noises.

We randomly sample robot poses, and for each robot pose, we randomly sample 5 different camera poses. In addition to pure random sampling, we also add some camera poses and robot poses similar to those in the RT-X datasets and add perturbations. We obtain paired images between different robots and their segmentation mask from Robosuite, and we add random brightness augmentation with range $[-40, 40]$ to the source robot images to increase the robustness of the segmentation model and R2R model to real-world lighting. In this way, we obtain about 800k images for each of the 4 robot types: Franka, UR5, Sawyer, and Jaco. See Fig. 5 for some example images.

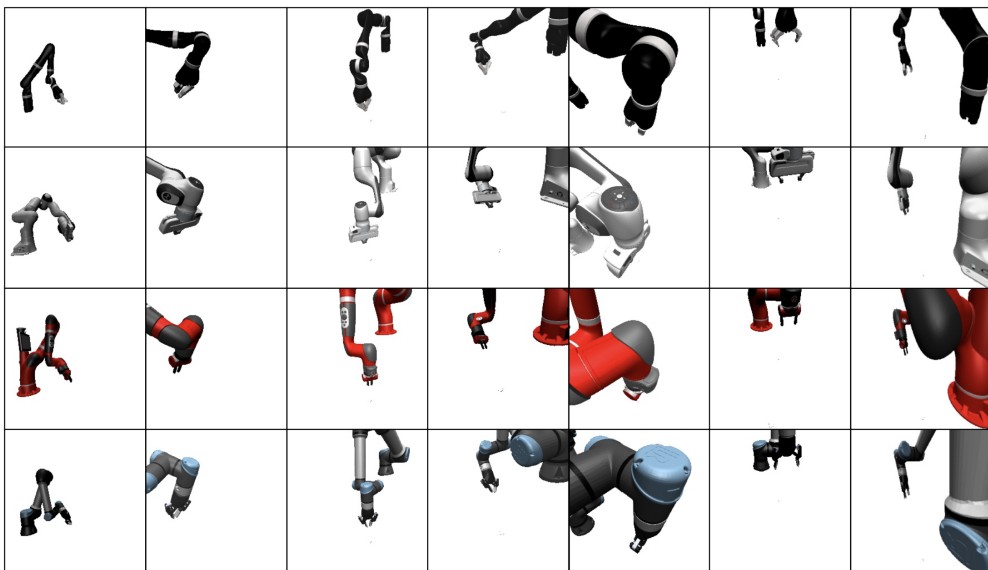

Figure 5: **Example of paired images for training the R2R model.** We use Robosuite [143] to generate pairs of Jaco, Franka, Sawyer, and UR5 at the same pose.

To create the dataset for training the segmentation model, we paste the generated robot image onto backgrounds from ImageNet [137]. See Fig. 6 for some example images.

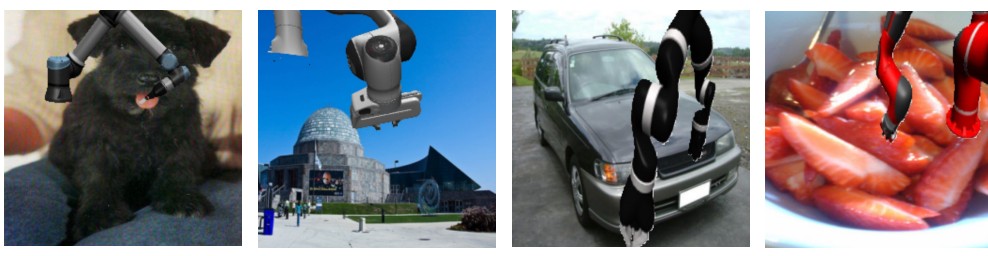

Figure 6: Example of pasted images on ImageNet used for training the segmentation model.

### 7.2.2 Model Training Details

Regarding the robot segmentation model, we fine-tune SAM with LoRA with 4 A6000 GPU for 1.5 hours. In particular, we leverage mixed-precision (8-bit and 16-bit) and the torch.compile feature to accelerate training. The model is trained with a mini-batch size of 64, a learning rate of 1e-5, and a LoRA rank of 4.

Regarding the Robot-to-Robot generation model, we finetune Stable Diffusion with ControlNet on 1 A100 GPU for 36 hours on 800K paired images. We use a learning rate of 1e-4 and a batch size of 512. During inference, we leverage the Stream Batch proposed by Kodaira et al. [147] to batchify the generation phase, making the generation phase achieve around 3.2 FPS.

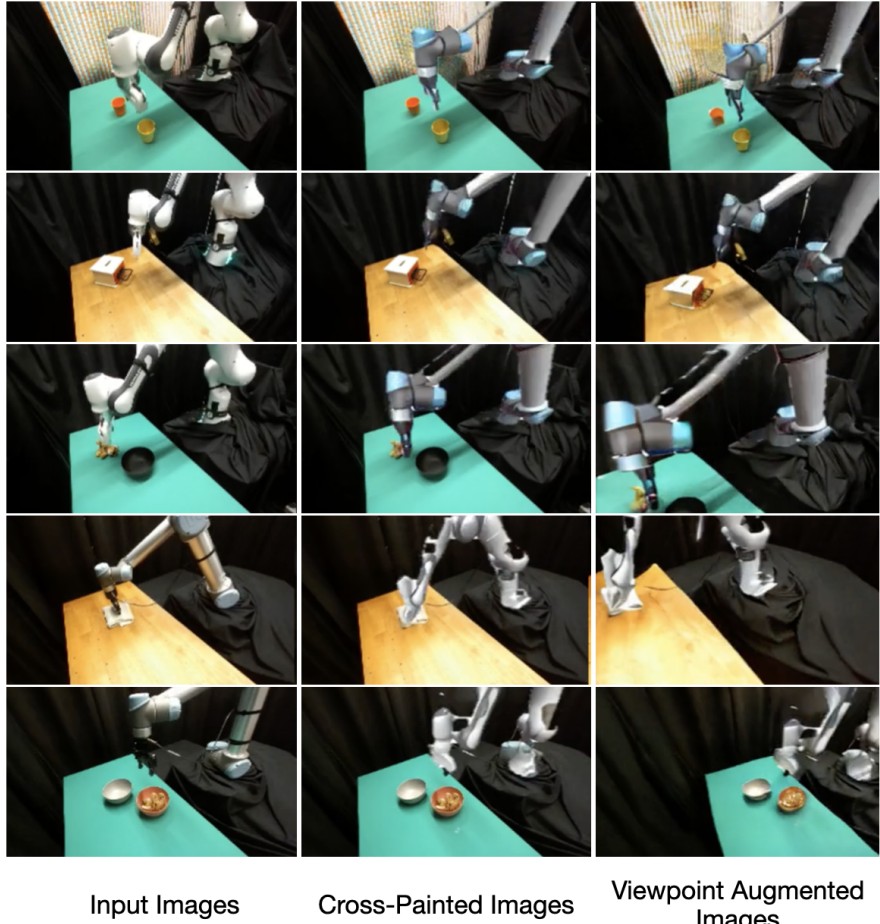

| Input Images | Cross-Painted Images | Viewpoint Augmented Images |

Figure 7: **Example of RoVi-Aug results.** We show some example results of RoVi-Aug applied to the training images of the 5 tasks.

We use ZeroNVS and the video inpainting model E2FGVI off-the-shelf without finetuning.

### 7.2.3 Computation Time for Data Augmentation

The advantage of RoVi-Aug over Mirage is that the primary of the compution is performed offline, not during execution time. Moreover, each model in RoVi-Aug's pipeline can be parallelized to process batchified video frames efficiently. We measured the throughputs of each module: Robot segmentation model achieves 4.1FPS, Robot-to-Robot achieves 3.2FPS, and the video inpainting model achieves 4.6FPS. On a single A100 GPU, it takes about 4-5 hours to perform Ro-Aug on a dataset of 200 trajectories. Similarly, the throughput for ZeroNVS inference is 1.3FPS, translating to 4.2 hours of viewpoint augmentation time on a dataset of 200 trajectories.

### 7.2.4 Example Augmented Images

In Fig. 7, we show some example results of RoVi-Aug applied to the training images of the 5 tasks. The left column is the original images; the middle column is the cross-painted images using the robot augmentation pipeline; the right column shows the view augmented images applied on top of the robot augmented images. The black regions in the generated robot are due to incomplete segmentation mask (missing some regions in the generated robot) when pasting the generated robot to the original image. We can see that in general, RoVi-Aug generates diverse view angles of the target robot performing the task of interest.

### 7.2.5 Generation Artifacts

We observe a few different types of artifacts: 1) illumination difference, 2) inaccurate object segmentation, 3) temporal inconsistency, and 4) inaccurate robot-to-robot generation.

For 1), since there are almost always differences in the lighting conditions between the simulated images that are used to train the R2R diffusion model and that of the test robots which are unknown a priori, we perform random brightness augmentation to the generated robot scenes in the augmentation pipeline. As shown in Table 1, we find this mitigation strategy is generally effective.

For 2), the robot segmentation model may sometimes under-segment or over-segment, particularly when the source robot is occluded or interacting with objects. As the R2R diffusion model is not trained on source robot images with objects in the gripper or with a partially segmented robot, the generated target robot can have large artifacts including distortion or hallucination due to out-of-distribution inputs.

For 3), due to the stochastic nature of diffusion models and possible multiple inverse kinematics solutions for putting the end effector of the target robot at the position of the source robot with different joint angles, the generated images may not be consistent across time. We did not observe this as a big problem potentially due to two reasons: (1) The Diffusion Policy does not use a long history so temporally inconsistent artifacts may not have a large effect; (2) The stochasticity of the generated images has an effect of randomization, which may help the policy be more robust to visual artifacts. Future work could also use a video diffusion model [13] to perform robot generation based on the entire robot trajectory to improve robot pose consistency.

For 4), even though our robot-to-robot diffusion model is trained on a large number of paired robot data, the generated images may still contain visible artifacts. For example, due to the ambiguity of inferring the field of view parameter from an image, the generated robot arm may be too thin or too thick. The generated gripper may also have artifacts or its position or orientation may not completely align with the source robot.

Due to these artifacts, we observe in Table 1 that Mirage achieves better performance than Ro-Aug on tasks that require more precision, such as cup stacking. This is because Mirage has the benefit of using a URDF with precise camera calibration to put the gripper at the exact location desired. On the other hand, artifacts in the R2R Generation model mean that the gripper of the target robot may not have the exact same pose as the original robot. However, as we show in Table 2, the ability of RoVi-Aug to perform finetuning can bring the performance higher than Mirage.

### 7.3 Physical Experiment Details

We provide more details on the physical experiment setups described in Section 5.2.

For the Franka-UR5 transfer experiments, we study 5 tasks: (1) Open a drawer, (2) Pick up a toy tiger from the table and put it into a bowl (Place Tiger), (3) Stack cups, (4) Sweep cloth from right to left, and (5) Transport a toy tiger between two bowls. For each task, the initial position of the robot gripper is randomized. For (1), the position and orientation of the drawer on the table is randomized, and the goal for the robot gripper is to go into the handle, pull it out, and leave the drawer. For (2), the positions of the tiger and the drawer are randomized. For (3), the positions of both cups are randomized. For (4), the initial position of the cloth is randomized in the right region of the table, and the robot needs to push it to the left region of the table, a distance of about 0.5 m. For (5), there are 2 bowls (red and grey) whose positions are randomized, and the toy tiger is always in the red bowl initially. The robot needs to grasp it and drop it into the grey bowl. Among them, stacking cup requires high precision and is most difficult, and sweeping cloth is the easiest.

For the OXE dataset experiments, the 2 tasks from the Berkeley UR5 datasets (Transport Tiger, Sweep Cloth) are the same as (4) and (5) above. For the 2 tasks from the Jaco Play datasets, the "Pick Cup" task requires the robot to pick up a cup that is randomly initialized on the table, and the "Bowl in Oven" task requires the robot to pick up a bowl and put it into a toaster oven.

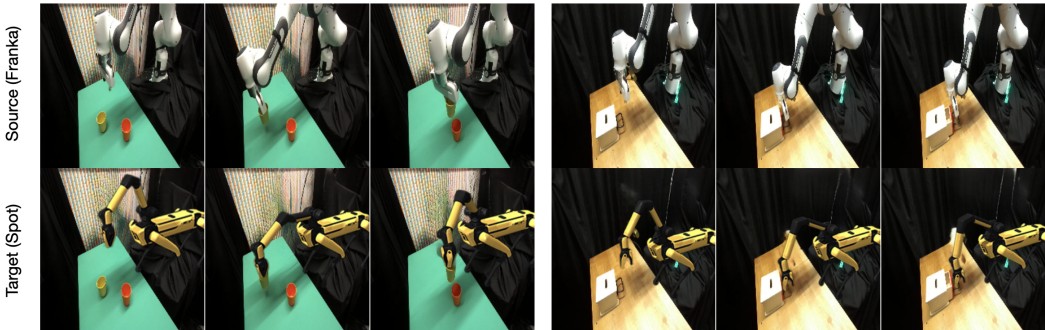

Figure 8: Example of robot augmentation from Franka to a Boston Dynamics Spot.

### 7.3.1 Policy Learning Details

We use the codebase from DROID [5] as our Diffusion Policy implementation, which is an open-source version integrated with Robomimic [148]. Similar to them, We use downsample camera observations at a resolution of $128 \times 128$ and the robot proprioception as input, and produce absolute robot end-effector translation, rotation, and gripper actions. And as with DROID and the original Diffusion Policy implementation, we train the diffusion policy to generate 16-step action sequences, and during rollouts, step 8 actions open loop before re-running policy inference. Compared to DROID, we use a ResNet-18 visual encoder instead of a pre-trained ResNet-50 for faster training, and we do not condition the policy with language input since we train a separate policy for each task (or 2 tasks for Table 5).

For few-shot finetuning experiments, we did not freeze any part of the diffusion policy and simply continued training on the target robot dataset (5/10 demonstrations) for only 100 epochs (about 20 minutes) to prevent the policy from overfitting to the target data too much.

### 7.3.2 Failure Modes

We describe the common failure modes of RoVi-Aug and baselines here. For the 3 pick and place tasks ("Place Tiger," "Stack Cup," and "Transport Tiger"), failure cases are usually missed grasp or inaccurate placing. For "Open Drawer," failure cases are typically gripper missing the drawer handle. For "Sweep Cloth," failure cases include inaccurate reaching and gripper being too high or leaving the table too early during the trajectory. For baselines, failure modes also include the robot getting confused and simply hovering over the objects without performing the task.

### 7.4 Model and Computation Details

For Ro-Aug, our segmentation model is a 636M-parameter SAM model with 35.6M-parameter LoRA layers; the video inpainting model E2FGVI is a 41.8M parameter model that we use off-the-shelf; the Robot-to-Robot (R2R) Generation model is a 1B-parameter Stable Diffusion model with around 350M-parameter ControlNet. For Vi-Aug, ZeroNVS is a 1B model that we use off-the-shelf. For policy learning, we use Diffusion Policy with a ResNet18 encoder and 1D-UNet with 80M parameters in total.

### 7.5 Example of Cross-Painting with a Mobile Robot

Generalization from arms mounted on a stationary base to mobile robots is much more challenging. In this section, we try an experiment using images of the Franka arm and apply robot augmentation to replace the Franka with a Boston Dynamics Spot to illustrate some examples with cross-painting to a mobile robot. While we do not have the hardware to perform physical experiments, the cross-painted images look somewhat realistic (see Figure 8), so it may be possible that the cross-painted Franka dataset could jumpstart the training for Spot. There are additional challenges associated with mobile

manipulation, such as coordination between base and arm movements and less accurate arm control, which we will leave as future work.

