# OpenReview forum: "RoVi-Aug: Robot and Viewpoint Augmentation for Cross-Embodiment Robot Learning"
_robot-learning.org/CoRL/2024/Conference — CoRL 2024_

### Official Review · Reviewer_scrS · 2024-07-15

**Originality:** 2
**Technical Quality:** 4
**Clarity Of Presentation:** 3
**Potential Impact:** 3
**Recommendation:** 3
**Confidence:** 4

**Review:**

Strength:

- Studies an important problem (transfer) and introduction motivates the problem well

- It is good to see the proposed method works on several real-world experiments!



Weakness:

- The citation format needs modification. Many published paper are cited as the arXiv preprint version. The author(s) should cite their published version.


- Zero-shot transfer performance on some tasks (e.g. Stack Cup task for Franka to UR5 transfer in Table 2) is lower than the Mirage baseline and the difference cannot be ignored.

- In Table 2, Why the performance of Ro-Aug is lower than that of the ablation method Ro-Aug w/o Bright. Rand? This result seems to contradict the conclusion that “The brightness randomization is crucial in preventing the policy from overfitting to the lighting in simulation that the R2R model is trained on” in the Results section.

- I cannot find a  pseudocode of the proposed algorithm in both the main text and the appendix. Providng clear pseudocode of it will be of great help.

- I would like to see the author(s) to report the computational resources used in their training process. How many GPUs/TPUs are used for the training process and how long it takes? In addition, I cannot find some other hyperparameters such as the batch size used for training. I wish to see this information in the section of Implementation Details or in the Appendix.

**Quality Of The Limitations Section:**

3

**Questions For Rebuttal:**

See the list of weaknesses in the section above.

**Robotics Focus:**

4

**Summary Of Paper:**

This paper proposes a data augmentation method for cross-embodiment transfer learning for robotic manipulation. They first finetune a Segment Anything Model (SAM) to segment the robot out of the image observation. Then use a ControlNet to finetune a pretrained Stable Diffusion to synthesize an alternative robot from a given robot. Lastly they use an inpainting model E2FGVI to inpaint the missing region of the background, and then paste the generated robot T back to the image. In addition,  ZeroNVS model is used to generate images from different viewpoints. These new images of different robots from different viewpoints are added into the original dataset for data augmentation in the training process. Experiments demonstrate good transfer performance to novel robot emboiments in the real world.

**Summary Of Recommendation:**

The paper is well motivated with a good set of experiments in the real world. Still writing has room for improvement (see list of weaknesses).

---

### Official Review · Reviewer_HKKU · 2024-07-20

**Originality:** 4
**Technical Quality:** 4
**Clarity Of Presentation:** 5
**Potential Impact:** 4
**Recommendation:** 4
**Confidence:** 4

**Review:**

### Strengths

* Approach: Proposes a novel approach for cross-robot embodiment using off-the-shelf vision models and fine-tuning with simulation data. And improves the viewpoint generalization for downstream tasks.
* Experiments: The proposed pipeline's effectiveness has been shown clearly with the real robot experiments. The difference to the closest baseline has been also explained and demonstrated successfully.

### Weaknesses
* The backgrounds used in the experimental setup are quite simplified. It’s unclear if this is due to the models' lack of capabilities for handling cluttered backgrounds. If so, this should be clearly stated in the limitations section. In the Appendix Figure 6, shows that robot images are placed on various backgrounds to robustify the model with respect to different backgrounds. Is this not enough to deploy the model with more complex backgrounds? What would it take to make this model work on not-so-simple backgrounds?

* I recommend explaining the compute budget used to generate the target robot data. The models used for the robot segmentation, robot-to-robot generation, robot inpainting, and view synthesis are large visual models and it would be useful for the readers to know how much computing is needed to reproduce the results. Also, it would be useful to know how long it takes to generate the 800k demonstrations for each robot. If one were to scale this method to tens of robots, would that be a scalable thing to do?

**Quality Of The Limitations Section:**

2

**Questions For Rebuttal:**

* While this paper presents extensive real-world results, some of the experiment's details are left out. For each result, how many runs are done? How long are the trajectories? What are the failure cases?

* The policy training details are not mentioned. Did you follow the exact setup from the original diffusion policy work? Please explain if any changes are made to the diffusion policy training. You can put these details in the Appendix.

* The policy fine-tuning details should be explained in Table 3. Did you freeze any part of the diffusion policy? Also, this table should include a result of co-training with the 5/10 demonstrations and augmented data.

* In Table 6, Octo-Base is trained with both robots and view augmentations. I’d strongly suggest that these results also show the models trained with only robots and only with viewpoint augmentation to see how much each method contributed to the final result.

**Robotics Focus:**

4

**Summary Of Paper:**

This paper addresses two major challenges in policy learning for manipulation: 1. Zero-shot cross-embodiment policy transfer 2. Generalization to novel viewpoints for visuomotor policies. The authors propose a vision pipeline that takes the demonstrations from the source robot rollouts and replaces the source robot with the target robot by processing them into a series of visual foundation models. First, they segment the robot mask, then transfer pixels of the source robot into the target robot pixels, and then place the transferred robot pixel onto the background image. With this augmentation dataset, they train a diffusion policy that can be deployed to the target robot zero shot. Furthermore, they use a viewpoint synthesis model to generate more data from different camera poses to make the model robust to the camera pose changes. They show the effectiveness of their method through a series of real-world experiments.

**Summary Of Recommendation:**

I recommend the acceptance of the paper since it proposes a novel visual pipeline for cross-body policy transfer and viewpoint generalization with strong real-world experiments.

---

### Official Review · Reviewer_XX5k · 2024-07-21
**RoVi-Aug: Robot and Viewpoint Augmentation for Cross-Embodiment Robot Learning**

**Originality:** 3
**Technical Quality:** 4
**Clarity Of Presentation:** 4
**Potential Impact:** 2
**Recommendation:** 3
**Confidence:** 4

**Review:**

Overall, this paper is very well-written, containing an abundance of citations, has a well-designed approach with each component explained, and has a structured evaluation with several positive results.


Strengths
+ This paper combines several techniques well to allow for better zero-shot transfer. The details of each component are explained well.
+ The results are positive with respect to the baseline, Mirage. The authors find that RoVi-Aug applied to OXE datasets improves policy performance by up to 30%
+ The authors do a good job of discussing the limitations of this framework.


Weaknesses
- An issue with utilizing diffusion models is artifacts in generated images. The authors should include further details about the generated images, whether image artifacts were present, and why they did not pose a problem.
- This framework seems limited to transfer to stationary arms. For example, I think there would be issues in transferring a robot policy demonstrated on a Franka arm to a Boston Dynamics spot equipped with an arm. Could you clarify if this is the case or if you feel this framework should be able to transfer well in this scenario?



Post Rebuttal - I thank the authors for responding to my questions and concerns. I will maintain my score of Weak Accept. Please include the information provided in the rebuttal responses in the final verison of the paper.

**Quality Of The Limitations Section:**

3

**Questions For Rebuttal:**

- Please address the weakness noted above.

Further Questions:
- How does the framework handle the mixture between different camera angles and different viewpoints of an object? For example, in the place tiger task, does the augmentation also need to generate different poses the toy tiger may assume?

**Robotics Focus:**

4

**Summary Of Paper:**

This paper presents RoVi-Aug, a framework for robot and viewpoint augmentation for cross-embodiment robot learning. This work builds on Mirage and addresses the key drawbacks of requiring precise robot models and camera matrices, not allowing fine-tuning, and inability to deal with camera perturbations. Utilizing diffusion models, the authors can generate trajectories with novel robots and viewpoints. Results display that RoVi-Aug outperforms prior work, increases robustness, and allows for zero-shot deployment to target robots.

**Summary Of Recommendation:**

Overall, this paper is very well-written, containing an abundance of citations, has a well-designed approach with each component explained, and has a structured evaluation with several positive results.  Many of the techniques that are combined are established but the authors do a good job of building up RoVi-Aug and explaining the need/training process for each component.

---

### Official Review · Reviewer_bj68 · 2024-07-30
**Review of Rovi-Aug**

**Originality:** 4
**Technical Quality:** 5
**Clarity Of Presentation:** 5
**Potential Impact:** 3
**Recommendation:** 4
**Confidence:** 4

**Review:**

Strength:

The paper addresses several issues present in an earlier work, Mirage. It eliminates the need to know the exact robot models and camera matrices and allows policy fine-tuning while also overperforming Mirage across different evaluations. The paper also shows the method's usefulness when used with methods such as Octo.

Weakness:
 - Without the video, it is hard to understand what happens in the experiments and what the performance metrics provided in the evaluation translate to.
 - It would be nice to see some details of the models such as model sizes
 - While the paper alleviates the need to know camera matrices and robot models of Mirage, the need to know the transformation matrix between the Cartesian spaces of the robots stands.
 - Even with the explanation in text, Table 4 is hard to understand
 - In line 268, there is a reference to Table 1 while there is a reference to Tables 2, 3 and 4 in the previous paragraphs. I understand that this is probably to make the paper fit to 8 pages, but this makes the paper hard to follow.
 - As stated in the limitations, system success depends heavily on the performance of the models it utilizes.
 - The evaluations heavily depend on the similarity between the robots’ embodiments. The arms’ morphology does not really matter but the gripper of Franka and the gripper of UR-5 (which looks like ROBOTIQ-85) are also too similar in shape and function. Hence, they can almost be directly mapped to each other.

**Quality Of The Limitations Section:**

3

**Questions For Rebuttal:**

- In line 210, you mentioned that you used 800k images for each robot. How does the model’s performance change when fewer or more images are used?
 - How many trials did you perform to get the results of each evaluation?
 - From what I understand, you map each pose of one robot to the other which works well since the robots are almost identical in how they function. However, how would the model perform if the embodiments were more different? For instance, instead of robotiq-85, the 3-finger adaptive gripper or BarrettHand was used. Theoretically, they could still achieve the same tasks without a problem, but since one of the grippers would not be symmetrical, the direct mapping would not translate to success in some tasks since they could not achieve them the same way. Please explain.

**Robotics Focus:**

4

**Summary Of Paper:**

The paper focuses on mitigating the issue of non-uniform distribution of robot types and camera view angles in robotic datasets to alleviate the problem of robotic policies overfitting to more common camera angles and robot embodiments. The paper includes extensive evaluations in real-world experiments, ablations using different settings, and comparisons to state-of-the-art methods.

**Summary Of Recommendation:**

The system utilizes state-of-the-art components and adresses several issues that were present in a prior method. The research questions and evaluations are also satisfactory.

---

### Author Rebuttal · Authors · 2024-08-12

We would like to thank all the reviewers for their thoughtful reviews as well as encouraging feedback. We are glad to see that the reviewers find the paper studies an important problem (Reviewer scrS), is well-written (Reviewer XX5k), the experiments extensive (Reviewer bj68), and that the proposed method is novel (Reviewer HKKU).

**We have incorporated the reviewers' feedback and updated our manuscript.** Specifically, we have clarified our assumptions, added more details about our experiment setup, evaluation, and computation resources, and discussed our limitations in more detail. We have also added two experiments requested by the reviewers. In the Appendix, we have added pseudocode, model training details including compute and hyperparameters, a discussion of artifacts and failure modes, and some example images of cross-painting from a Franka to a Boston Dynamics Spot robot.

We appreciate all the feedback and look forward to continued discussion with the reviewers about our updated manuscript.

---

### Decision · Program_Chairs · 2024-09-04

**Decision:**

Accept

**Comment:**

The paper focuses on mitigating the issue of non-uniform distribution of robot types and camera view angles in robotic datasets to alleviate the problem of robotic policies overfitting to more common camera angles and robot embodiments. This paper deals with a challenging and important problem in robotics, addresses the limitations of the previous Mirage method, outperforms the baseline, provides real-world experiments, and well-discusses the limitations.
The reviewers originally had important comments and concerns on the left-out details of the method and experiments, the similarity of the robot embodiments, the possible problems with diffusion models, and the left-out computational requirements of the method.
The reviewers now indicate that their concerns have been addressed by the author's revisions.
Therefore I recommend this paper accept as an oral presentation.